# Lamb Waves Propagation Characteristics in Functionally Graded Sandwich Plates

**DOI:** 10.3390/s22114052

**Published:** 2022-05-27

**Authors:** Jie Gao, Jianbo Zhang, Yan Lyu, Guorong Song, Cunfu He

**Affiliations:** 1Faculty of Materials and Manufacturing, Beijing University of Technology, Beijing 100124, China; jiegao@emails.bjut.edu.cn (J.G.); grsong@bjut.edu.cn (G.S.); hecunfu@bjut.edu.cn (C.H.); 2The 3th Research Institute of China Electronics Technology Group Corporation, Beijing 100015, China; zhangjianbo@emails.bjut.edu.cn

**Keywords:** functionally graded materials, Legendre polynomial series expansion method, Lamb wave dispersion curve, volume fraction curve, finite element simulation

## Abstract

Functionally graded materials (FGM) have received extensive attention in recent years due to their excellent mechanical properties. In this research, the theoretical process of calculating the propagation characteristics of Lamb waves in FGM sandwich plates is deduced by combining the FGM volume fraction curve and Legendre polynomial series expansion method. In this proposed method, the FGM plate does not have to be sliced into multiple layers. Numerical results are given in detail, and the Lamb wave dispersion curves are extracted. For comparison, the Lamb wave dispersion curve of the sliced layer model for the FGM sandwich plate is obtained by the global matrix method. Meanwhile, the FGM sandwich plate was subjected to finite element simulation, also based on the layered-plate model. The acoustic characteristics detection experiment was performed by simulation through a defocusing measurement. Thus, the Lamb wave dispersion curves were obtained by *V*(*f*, *z*) analysis. Finally, the influence of the change in the gradient function on the Lamb wave dispersion curves will be discussed.

## 1. Introduction

Functionally graded materials (FGM) are based on computer-aided material design, using advanced material compounding technology to make the elements (composition, structure, etc.) of the constituent materials continuously change from one side to the other along the thickness direction. Thus, the properties and functions of the material also vary in gradient. Functional gradient materials of metal-ceramics were proposed and prepared in 1984 [1]. Since the volume content of the FGM components is continuously changed in the spatial position, and there is no sudden change in physical properties, the interlayer stress problem can be avoided and the stress concentration phenomenon can be reduced. At the same time, FGM is a good devisable material, in which one can change the spatial distribution of composition and content of the material by a target function, so as to achieve the purpose of optimizing the internal stress distribution of the structure [2].

The FGM sandwich plate consists of three layers: the top layer, the middle layer and the bottom layer. Generally, FGM sandwich plates are divided into two categories. One is FGM as the top and bottom layers of the sandwich plate, and the homogeneous isotropic materials as the intermediate layer. The other type is FGM as the middle layer of the sandwich plate, and homogeneous isotropic materials as the top and bottom layers. FGM sandwich plates have excellent overall performance, and have been used in optical, biomedical, electromagnetic and mechanical engineering, etc. [3].

The elastic waves in the FGM sandwich plate contain ultrasound guided waves and body waves. Ultrasonic guided waves cover Lamb waves, surface waves, Love waves, etc. Ultrasonic guided waves provide unique capabilities for the structural health monitoring of plate-like structures [4]. However, the guided waves have multi-mode and dispersion characteristics during propagation, and the dispersion appears to be a unique physical property. It mainly indicates that the propagation characteristics of the guided waves are affected by frequency. That is to say, the propagation velocity of a guided wave will change by frequency, which is called dispersion [5]. In addition, most guided wave modes have strong dispersion characteristics. Therefore, studying the relationship between the dispersion curve of FGM sandwich plates and material property parameters is an important part of theoretical research. Zhu et al. [6] used the matrix recursion method to establish the characteristic equations of Lamb waves of multi-layer free plates, and analyzed the dispersion characteristics of double-layer plates and sandwich plates. Wu et al. [7] studied the propagation dispersion characteristics of Lamb waves from single-layer plates to multi-layer FGM plates, and obtained the relationship between the continuous change in material properties and the Lamb wave velocity and displacement. Bruck [8] analyzed the propagation of stress waves in FGM by establishing a one-dimensional FGM model, and transitioned the FGM layered model to a continuously changing gradient model. Chen et al. [9] used a layered plate model to analyze the dispersion characteristics of FGM plates under large frequencies and thick product conditions. In all the above research, the FGMs were divided into many homogeneous or inhomogeneous layers, in order to solve the wave propagation problem. However, the layer number of FGMs plays a vital role in the numerical accuracy of the calculations.

In addition, Lefebvre et al. [10] proposed the Legendre orthogonal polynomial series expansion (LOPSE) method to study the propagation properties of waves in layered-plate structures. Yu et al. [11] further introduced the Legendre series expansion method into the dispersion curve calculation of an anisotropic multilayer piezoelectric material plate with a greater difference in mechanical parameters. Compared with the rotation matrix method, a good calculation result is obtained. Dong et al. [12] studied the SH surface wave in the piezoelectric gradient half space, considering the horizontal shear direction displacement by using Laguerre orthogonal polynomials. Salah et al. [13] proposed a layered model to analyze the Love wave over a half space of an elastic substrate covered by a functionally gradient piezoelectric material plate. As mentioned above, the studies treated the FGM structures as a continuously gradient medium, and they effectively calculated the propagation characteristics of acoustic waves in FGMs without separating them into multilayer plates. However, there are few reports on the numerical simulation of Lamb wave propagation in FGM sandwich structures.

Likewise, the finite element method is a numerical method with both a theoretical basis and practical significance. It was originally used by Zienkiewicz [14] to simulate wave propagation and scattering, but then Finnveden [15] successively used the spectral finite element method to study the periodic waveguide structure and the guided wave in the viscoelastic damped waveguide structure. Cheng et al. [16] studied the propagation of surface acoustic waves excited by lasers in functionally graded materials, and simulated the gradients of various mechanical and thermal parameters in functionally graded materials. Kim and Paulino [17] proposed an isoparametric gradient element model, and applied the shape function of the model to obtain the material properties of the attribute of the element node to the inner difference. Zhang and Xiao [18] applied this method to prove that the finite element model based on isoparametric gradient elements can better reflect the gradient variation in material properties. Wang and Gross [19] proposed a layered model of FGM. The material parameters of each layer change according to a continuous function and are continuous at the interface. Such a layered model achieved good results in the crack analysis of FGM structures. Nevertheless, little research paid attention to the complex multi-mode dispersion characteristics of functionally graded materials, which can provide more abundant information for non-destructive testing and the evaluation of the characteristics of FGM plates.

In this research, we use the Legendre polynomial series expansion method to study the propagation of Lamb waves in functionally graded material sandwich plates, and discuss their convergence problem. The influence of gradient layer parameter changes on Lamb wave dispersion curves will also be given. In addition, the finite element model of FGM sandwich plates was established by PZFlex (Division of Applied Science, Mountain View, United States), and the experimental process of defocusing the measurement of line-focused ultrasonic transducers based on acoustic microscopy, also known as the *V*(*f*, *z*) measurement, was simulated.

## 2. Theoretical Derivation and Numerical Results

### 2.1. Modeling

For a functionally graded sandwich plate, as shown in Figure 1, the propagation direction of the Lamb wave is along the *x*_1_ axis. The thickness of the sandwich plate is *h*_1_ + *h*_2_ + *h*_3_, in which *h*_2_ is the thickness of the FGM layer, and *h*_1_ and *h*_3_ are the thicknesses of steel and copper, respectively. The material parameters of the FGM layer vary continuously in the thickness direction. Here, we are referring to the density and the elastic constants, which are functions of *x*_3_.

Assuming that the displacement components of the Lamb wave are the following:(1)$$\begin{array}{l} u_{1}=U\left(x_{1},x_{3},t\right)\\ u_{2}=0\\ u_{3}=W\left(x_{1},x_{3},t\right) \end{array}$$
then the equations of motion will be given as follows:(2)$$\begin{array}{l} \frac{\partial \sigma_{11}}{\partial x_{1}}+\frac{\partial \sigma_{13}}{\partial x_{3}}=\rho \frac{\partial^{2}u_{1}}{\partial t^{2}}\\ \frac{\partial \sigma_{31}}{\partial x_{1}}+\frac{\partial \sigma_{33}}{\partial x_{3}}=\rho \frac{\partial^{2}u_{3}}{\partial t^{2}} \end{array}$$

Geometric relationship under the assumption of small deformation is as follows:(3)$$\varepsilon_{ij}=\frac{1}{2}\left(\frac{\partial u_{i}}{\partial x_{j}}+\frac{\partial u_{j}}{\partial x_{i}}\right)\;\;\;\;\;\left(i,j=1,2,3\right)$$

Free harmonics of the particle displacement can be written as follows:(4)$$\begin{array}{l} u_{1}=U(x_{3})\cdot e^{\left[i(kx_{1}-\omega t)\right]}\\ u_{3}=W(x_{3})\cdot e^{\left[i(kx_{1}-\omega t)\right]} \end{array}$$
where *σ_ij_* and *ε_ij_* represent stress and strain, respectively, *U*(*x*_3_) and *W*(*x*_3_) are the amplitudes of particle vibrations on the *x*_1_ and *x*_3_ direction, *k* is the wave number, and *ω* is the angular frequency.

Considering the boundary problem of isotropic plates, the rectangular window function can be introduced by the following:(5)$$\pi_{h_{n}}(x_{3})=\{\begin{array}{l} 1,\;0\leq x_{3}\leq h_{1}+h_{2}+h_{3}\\ 0,\;\;\;\;\;elsewhere \end{array}$$

The elastic constant and density of the material are expressed as a function of position, as follows:(6)$$\begin{array}{l} C_{ij}=\sum_{n=1}^{N}C_{ij}^{n}\pi_{h_{n}}(x_{3})\\ \rho =\sum_{n=1}^{N}\rho^{n}\pi_{h_{n}}(x_{3}) \end{array}$$
where *N* is the total number of layers, and here, *N* = 3. Therefore, the elastic constants and density in the sandwich plate can be expressed as follows:(7)$$\begin{array}{c} C_{ij}(x_{3})=C_{ij}^{1}\pi_{h_{1}}(x_{3})+C_{ij}^{2}(x_{3})\pi_{h_{2}}(x_{3})+C_{ij}^{3}\pi_{h_{3}}(x_{3})\\ \rho (x_{3})=\rho^{1}\pi_{h_{1}}(x_{3})+\rho^{2}(x_{3})\pi_{h_{2}}(x_{3})+\rho^{3}\pi_{h_{3}}(x_{3}) \end{array}$$

The middle layer of the sandwich plate is the FGM layer; the volume fraction of copper of this layer is represented as *V*_Cu_, which can be written by a power function, as follows:(8)$$V_{\mathrm{Cu}}={\left(1-\frac{x_{3}-h_{1}}{h_{2}}\right)}^{n}\;\;\left(h_{1}\leq x_{3}\leq h_{1}+h_{2},\;0\leq n\leq \infty \right)$$
where *n* is the exponent of the power function. The propagation characteristics of Lamb waves in the FGM layer under different gradient distributions can be obtained by changing the power exponent *n*. Then, in the FGM layer, the relationships between elastic constants/density and volume fraction are as follows:(9)$$\begin{array}{l} C_{IJ}^{\mathrm{steel}}=C_{IJ}^{\mathrm{steel}}+\left(C_{IJ}^{\mathrm{Cu}}-C_{IJ}^{\mathrm{steel}}\right)V_{\mathrm{Cu}}\\ \rho^{\mathrm{steel}}=\rho^{\mathrm{steel}}+\left(\rho^{\mathrm{Cu}}-\rho^{\mathrm{steel}}\right)V_{\mathrm{Cu}} \end{array}$$

Substituting Equation (8) into Equation (9) yields the functions of the elastic constant and density in the FGM layer, with respect to *x*_3_:(10)$$\begin{array}{l} C_{IJ}^{\mathrm{steel}}\left(x_{3}\right)=C_{IJ}^{\mathrm{steel}}+\left(C_{IJ}^{\mathrm{Cu}}-C_{IJ}^{\mathrm{steel}}\right){\left(1-\frac{x_{3}-h_{1}}{h_{2}}\right)}^{n}\\ \rho^{\mathrm{steel}}\left(x_{3}\right)=\rho^{\mathrm{steel}}+\left(\rho^{\mathrm{Cu}}-\rho^{\mathrm{steel}}\right){\left(1-\frac{x_{3}-h_{1}}{h_{2}}\right)}^{n} \end{array}$$

Thus, the constitutive relationship is given as follows:(11)$$\begin{array}{l} \sigma_{11}=\left[C_{11}\left(x_{3}\right)\varepsilon_{11}+C_{13}\left(x_{3}\right)\varepsilon_{33}\right]\cdot \pi \left(x_{3}\right)\\ \sigma_{33}=\left[C_{13}\left(x_{3}\right)\varepsilon_{11}+C_{33}\left(x_{3}\right)\varepsilon_{33}\right]\cdot \pi \left(x_{3}\right)\\ \sigma_{13}=2C_{55}\left(x_{3}\right)\varepsilon_{13}\cdot \pi \left(x_{3}\right) \end{array}$$

### 2.2. Legendre Orthogonal Polynomial Expansion

Substituting Equations (3), (4), (7), (10) and (11) into Equation (2) yields the wave control equation in the *x*_1_–*x*_3_ plane. Then, the wave control equation in the *x*_1_ direction is as follows:(12)$$\begin{array}{l} \left[C_{11}^{1}\pi_{h_{1}}(x_{3})+C_{11}^{2}(x_{3})\pi_{h_{2}}(x_{3})+C_{11}^{3}\pi_{h_{3}}(x_{3})\right]\cdot i^{2}k^{2}\cdot U\\ +\left[C_{13}^{1}\pi_{h_{1}}(x_{3})+C_{13}^{2}(x_{3})\pi_{h_{2}}(x_{3})+C_{13}^{3}\pi_{h_{3}}(x_{3})\right]\cdot ik\cdot W^{\prime }\\ +C_{55}^{2}{}^{\prime }(x_{3})\pi_{h_{2}}(x_{3})\cdot U^{\prime }+\left[C_{55}^{1}\pi_{h_{1}}(x_{3})+C_{55}^{2}(x_{3})\pi_{h_{2}}(x_{3})+C_{55}^{3}\pi_{h_{3}}(x_{3})\right]\cdot U^{\prime \prime }\\ +C_{55}^{2}{}^{\prime }(x_{3})\pi_{h_{2}}(x_{3})\cdot ik\cdot W+\left[C_{55}^{1}\pi_{h_{1}}(x_{3})+C_{55}^{2}(x_{3})\pi_{h_{2}}(x_{3})+C_{55}^{3}\pi_{h_{3}}(x_{3})\right]\cdot ik\cdot W^{\prime }\\ +\left[\begin{array}{l} C_{55}^{1}[\delta (x_{3}-0)-\delta (x_{3}-h_{1})\;]+C_{55}^{2}(x_{3})[\delta (x_{3}-h_{1})-\delta (x_{3}-h_{1}-h_{2})\;]\\ +C_{55}^{3}[\delta (x_{3}-h_{1}-h_{2})-\delta (x_{3}-h_{1}-h_{2}-h_{3})\;] \end{array}\right]\cdot \left(U^{\prime }+ik\cdot W\right)\\ =-\left[\rho^{1}\pi_{h_{1}}(x_{3})+\rho^{2}(x_{3})\pi_{h_{2}}(x_{3})+\rho^{3}\pi_{h_{3}}(x_{3})\right]\cdot \omega^{2}\cdot U \end{array}$$

The wave control equation in the *x*_3_ direction is as follows:(13)$$\begin{array}{l} \left[C_{55}^{1}\pi_{h_{1}}(x_{3})+C_{55}^{2}(x_{3})\pi_{h_{2}}(x_{3})+C_{55}^{3}\pi_{h_{3}}(x_{3})\right]\cdot ik\cdot U^{\prime }\\ +\left[C_{55}^{1}\pi_{h_{1}}(x_{3})+C_{55}^{2}(x_{3})\pi_{h_{2}}(x_{3})+C_{55}^{3}\pi_{h_{3}}(x_{3})\right]\cdot i^{2}k^{2}\cdot W\\ +C_{13}^{2}{}^{\prime }(x_{3})\pi_{h_{2}}(x_{3})\cdot ik\cdot U+\left[C_{13}^{1}\pi_{h_{1}}(x_{3})+C_{13}^{2}(x_{3})\pi_{h_{2}}(x_{3})+C_{13}^{3}\pi_{h_{3}}(x_{3})\right]\cdot ik\cdot U^{\prime }\\ +C_{33}^{2}{}^{\prime }(x_{3})\pi_{h_{2}}(x_{3})\cdot W^{\prime }+\left[C_{33}^{1}\pi_{h_{1}}(x_{3})+C_{33}^{2}(x_{3})\pi_{h_{2}}(x_{3})+C_{33}^{3}\pi_{h_{3}}(x_{3})\right]\cdot W^{\prime \prime }\\ +\left[\begin{array}{l} C_{13}^{1}[\delta (x_{3}-0)-\delta (x_{3}-h_{1})\;]+C_{13}^{2}(x_{3})[\delta (x_{3}-h_{1})-\delta (x_{3}-h_{1}-h_{2})\;]\\ +C_{13}^{3}[\delta (x_{3}-h_{1}-h_{2})-\delta (x_{3}-h_{1}-h_{2}-h_{3})\;] \end{array}\right]\cdot ik\cdot U\\ +\left[\begin{array}{l} C_{33}^{1}[\delta (x_{3}-0)-\delta (x_{3}-h_{1})\;]+C_{33}^{2}(x_{3})[\delta (x_{3}-h_{1})-\delta (x_{3}-h_{1}-h_{2})\;]\\ +C_{33}^{3}[\delta (x_{3}-h_{1}-h_{2})-\delta (x_{3}-h_{1}-h_{2}-h_{3})\;] \end{array}\right]\cdot W^{\prime }\\ =-\left[\rho^{1}\pi_{h_{1}}(x_{3})+\rho^{2}(x_{3})\pi_{h_{2}}(x_{3})+\rho^{3}\pi_{h_{3}}(x_{3})\right]\cdot \omega^{2}\cdot W \end{array}$$

The amplitudes of *U*(*x*_3_) and *W*(*x*_3_) of the displacements are expanded into the form of a summation of the Legendre orthogonal polynomials, which can be written as follows:(14)$$\begin{array}{l} U(x_{3})=\sum_{m=0}^{\infty }p_{m}^{1}\cdot Q_{m}(x_{3})\\ W(x_{3})=\sum_{m=0}^{\infty }p_{m}^{3}\cdot Q_{m}(x_{3}) \end{array}$$
where (*i* = 1, 3; *m* = 1, 2, …, *M*) are the expansion coefficients of *Qm*(*x*_3_). Theoretically, m takes from zero to infinity, but, in fact, m takes a finite value of *M*. Higher-order terms can be considered as infinitesimal quantities, and *M* is the cutoff order of Legendre orthogonal polynomial series. It should be noted that *Qm*(*x*_3_) is an orthogonally normalized polynomial group, as follows:(15)$$Q_{m}(x_{3})=\sqrt{\frac{2m+1}{h_{1}+h_{2}+h_{3}}}P_{m}(\frac{2x_{3}-h_{1}-h_{2}-h_{3}}{h_{1}+h_{2}+h_{3}})$$

Substituting the displacement amplitude Equation (14) into the wave control Equations (12) and (13) will derive the final form of Legendre polynomial expansion equations. Multiply both sides of the expanded equation by *Q_j_*(*x*_3_) and integrate *x*_3_ from zero to *h*_1_ + *h*_2_ + *h*_3_. Using the orthogonal properties of the Legendre polynomial, a matrix form of the equations can be given, as follows:(16)$$\left[\begin{array}{cc} A_{11}^{j,m} & A_{12}^{j,m}\\ A_{21}^{j,m} & A_{22}^{j,m} \end{array}\right]\left\{\begin{array}{c} p_{m}^{1}\\ p_{m}^{3} \end{array}\right\}=-\omega^{2}\left[\begin{array}{cc} M_{m}^{j} & 0\\ 0 & M_{m}^{j} \end{array}\right]\left\{\begin{array}{c} p_{m}^{1}\\ p_{m}^{3} \end{array}\right\}$$
where $A_{ij}^{j,m}$ and $M_{m}^{j}$ can be obtained from the wave control equations after expansion, which are shown in Appendix A. According to the matrix Equation (16), the relationship between the wave number *k* and the angular frequency ω can be obtained by solving the eigenvalues. That is how the dispersion curves of the Lamb wave in the FGM sandwich plate can be extracted.

### 2.3. Numerical Results and Discussion

#### 2.3.1. Convergence Analysis of Cutoff Order M

The material selected was a copper–FGM–steel sandwich plate, and the mechanical performance parameters of copper and steel are shown in Table 1. In total, the thickness of the plate is 0.4 mm, in which both the thicknesses of copper and steel are 0.1 mm, and the thickness of the FGM layer is 0.2 mm. According to Equation (8), the volume fraction of copper in the FGM layer along the thickness direction will take the indices *n* = 0.2, 0.5, 1, 2, 10, respectively, as illustrated in Figure 2.

From the LOPSE method, it can be concluded that in the process of solving the Lamb wave dispersion curves, once the number of polynomials exceeds a certain threshold, the phase velocity will infinitely approach the eigenvalue. Calculations of the Lamb wave dispersion curves in the frequency range of 0–10 MHz under seven cutoff orders (*M* = 3, 4, 5, 6, 7, 8, 9) are shown in Figure 3a–g, where the volume fraction index is *n* = 0.2. It can be observed that as the cutoff order *M* increases, the Lamb wave dispersion curve shows a convergence trend, which is consistent with the characteristics of the LOPSE method. This verifies the feasibility of the theoretical method. As can be observed from Figure 3h, when *M* = 8 and *M* = 9, the curves are substantially coincident, so the Lamb wave dispersion curves obtained at *M* = 8 are taken as a convergence solution.

Taking the volume fraction curve at *n* = 0.2 as an example, according to Equation (10), the elastic constant *C_IJ_* and density *ρ* of the FGM layer can be sliced into 10 equal minor sub layers. Meanwhile, when *N* = 1, the corresponding material layer is Cu; and when *N* = 10, the corresponding material layer is steel, and the material parameters can be obtained from Table 1. The thickness of each layer is 0.02 mm, and the material parameters vary in the same step. The corresponding parameters of all layers can be obtained from Equation (10), which are shown in Table 2.

The parameters from Table 2 are used to obtain a comparison result from Disperse (Imperial College NDT Laboratory, London, UK), also under *n* = 0.2. Compared with the convergent solution from the Legendre orthogonal polynomial expansion method, the results are shown in Figure 4. It can be observed that the curve is basically consistent in the frequency range of 0–10 MHz, which means that the theoretical solutions of the non-sliced model by Legendre expansion are consistent with the solutions of the sliced model by Disperse. In this case, *M* = 8 will approach closely enough to the results of the global matrix method [21].

#### 2.3.2. Effect of Volume Fraction *n* on Dispersion Curves

Under different power exponents, the gradient distribution of material parameters in the FGM layer is different, which has a certain influence on the Lamb wave dispersion curves. The dispersion curves of Lamb waves in the sandwich plate under different gradient distributions are calculated, as shown in Figure 5. The cutoff order of the Legendre orthogonal polynomial is also *M* = 8. Figure 5a–e show Lamb wave dispersion curves in five different sandwich plates, with *n* = 0.5, 1, 5, 10, 20, respectively. It can be observed that as the power exponent increases, the phase velocity of *S*0 mode at a low frequency range gradually increases. Meanwhile, the same phenomenon shows up in the higher-order modes. According to Figure 5, when the power index is gradually increased to infinity, the copper content in the FGM layer almost reduces to zero, and the sandwich plates can be regarded as double-layered plates with a top layer of 0.1 mm copper and a bottom layer of 0.3 mm steel. The Lamb wave dispersion curve in the copper–steel double-layered plate calculated by the Legendre orthogonal polynomial method is shown in Figure 5f.

#### 2.3.3. Displacement and Stress Distribution

The amplitude distribution of displacements and stress components along the thickness direction is the wave structure. According to the calculation result of the dispersion curve at *n* = 1 in Figure 5b, the eigenvector and its corresponding eigenvalue are calculated. Then, the displacement distribution of the different Lamb wave modes at different frequencies can be obtained. An arbitrary frequency *f* = 2 MHz is selected, and the Lamb wave velocities corresponding to the A0 (anti-symmetric zero-order mode) and S0 (symmetrical zero-order mode) modes at this frequency are 1933 m/s and 4469 m/s, respectively. The matrix eigenvectors $p_{m}^{1}$ and $p_{m}^{3}$ are inversely obtained by using the angular frequency ω corresponding to the two wave velocities as the eigenvalues. Substituting $p_{m}^{1}$ and $p_{m}^{3}$ into the Equation (16), the displacement and stress distribution in the FGM sandwich plate can be obtained, as shown in Figure 6 and Figure 7. So, the displacement and stress distribution curves corresponding to the arbitrary modes of the Lamb wave at any frequency can be obtained. 

It can be observed from Figure 6 that, with the gradual change in the material composition in the FGM sandwich plate along the thickness direction, the LOPSE method can ensure that the displacement variation in the plate is continuous. Additionally, due to the gradual change in the material composition, its displacement distribution no longer has a strict “symmetric” or “asymmetric” distribution, with respect to the center position of the plate. The advantage of the LOPSE method is that the sandwich plate can be calculated globally without delamination, thus solving the problem of stress discontinuity at the boundary. In the calculation, the stress distribution of the Lamb wave can be obtained by simply substituting the obtained displacement solution into the constitutive equation and the geometric equation. As can be observed from Figure 7, the stress components *σ*31 and *σ*33 are continuously distributed in the FGM sandwich plate, and the stress components at the top and bottom boundaries are zero.

## 3. Finite Element Analysis

### 3.1. Simulation Model

Based on ultrasonic microscope technology, an acoustic measurement simulation model with an FGM sandwich plate was established, and the corresponding Lamb wave dispersion curve was extracted. For the functionally graded material sandwich panel, the thickness of the sandwich plate is *h*_1_ + *h*_2_ + *h*_3_, in which *h*_2_ is the thickness of the FGM layer, and *h*_1_ and *h*_3_ are the thicknesses of steel and copper, respectively. In order to simulate the structural characteristics of nonhomogeneous materials (FGM layer), the corresponding material properties should vary between homogeneous steel and copper. Meanwhile, it is assumed that the material properties of each element layer are constant, and the material properties mesh uniformly along the thickness direction [22,23]. A number of subdivisions can approximate the continuous property variation; the corresponding propagation characteristics of acoustic waves are close to the graded type at this time [24]. On the other hand, when using the commercial finite element package PZFlex to simulate the distribution of sound field in materials, it is very important to assign mechanical property parameters to the corresponding layer of the FGM sandwich plate. In this problem, the uniform element with a thickness of 0.02 mm can solve the numerical simulation of sound field distribution for functionally graded material layers with a thickness of 0.2 mm.

In this section, a two-dimensional finite element model for a line-focusing ultrasound transducer was built in PZFlex. The dimensional parameters and material properties of the finite element model of the line-focusing ultrasonic transducer were referred to with the ultrasonic transducer used in the experiment. In the model, a piezoelectric polymer of polyvinylidene fluoride (PVDF) film was selected as the excitation/receiving element, and the polarization direction is directed to the center of the circle. The upper surface of the film is the positive electrode and the lower surface is the negative electrode. Back10 (tungsten-loaded epoxy, 10% VF, 5.8 Mray1) was used as the backing. Water was selected as the coupling medium for detection, and a copper–FGM–steel sandwich plate was used as the specimen.

Table 3 shows the material parameters of the model. The top layer of the specimen is copper, the middle is layered FGM, and the bottom is steel. The transverse/longitudinal wave velocity and density of copper and steel are known. The material parameters of the FGM layered model are obtained from the volume fraction curve (*n* = 0.2). The parameters of each layer are shown in Table 3. The thickness, focus radius, and full opening angle of PVDF film were set to 40 μm, 20 mm, and 80°, respectively. Then, this finite element model can be simplified to a two-dimensional model, as shown in Figure 8. The signal excited by the line-focusing ultrasonic transducer is a transient wide-band signal. Therefore, the excitation signal in the simulation selects the Sine-Impulse broadband signal with a central frequency of 7 MHz.

The simulation started at the focusing plane. Generally, at around 28 μs, the PVDF film receives the reflected echo from the bottom surface of the specimen for the first time. Thus, in this simulation, the propagating times of the acoustic waves were set to 35 μs. The finite element model is discretized by a rectangular grid, and a unit wavelength is divided by 20 grid nodes in water. It should be noted that the bottom surface of the model is set as a free boundary. In order to prevent reflection, the other boundaries of the model are set as absorbing boundaries.

### 3.2. Simulation Results

By changing the relative position of the ultrasonic transducer to achieve equal interval defocusing, a defocusing measurement simulation based on an ultrasonic microscopy technique was simulated, which is called *V*(*f*, *z*) analysis [25]. The defocus distance was 15 mm and the step was 0.025 mm. The finite element simulation was performed on each defocus position, and, in total, 600 sets of simulation data were obtained. The Lamb wave dispersion curve can be extracted by performing 2D Fourier transform of the time and space domains, as shown in Figure 9.

The Lamb wave dispersion curves from the simulation were superimposed with the dispersion curves from the LOPSE method, as shown in Figure 10. It can be observed from the figure that the theoretical results solved by the LOPSE method using the volume fraction index are consistent with the finite element simulation results using the layered model. Therefore, this result lays the theoretical foundation for FGM characterization by acoustic microscopy.

## 4. Conclusions

In this research, the problem of Lamb wave propagation in the FGM sandwich plate without discretizing the gradient structure into a homogeneous multilayered model is solved numerically.

(1) The LOPSE method is employed for solving the Lamb wave dispersion curves and their displacement and stress distributions, even when the material parameters vary continuously along the thickness direction. The convergence of the results by a polynomial method is analyzed, and the convergence solution is also obtained. Moreover, the convergence solution is basically consistent with the results calculated using the global matrix method.

(2) The middle layer of the sandwich plate is FGM, in which the material parameter changes gradiently along the thickness direction. By solving the Lamb wave dispersion curve of the sandwich plate under different gradient distributions, it is obvious that the volume fraction of the top layer material in the FGM layer decreases and the volume fraction of the underneath layer material increases when the power exponent increases, then the dispersion relation of the Lamb wave gradually approaches a double-layer plate.

(3) The finite element model of the FGM sandwich plate is established by slicing the FGM into layers, and the defocus measurement simulation by a line-focusing ultrasonic transducer was carried out based on an acoustic microscopy technique. The extracted Lamb wave dispersion curves are basically consistent with the theoretical calculation results, which further verifies the LOPSE method. Then, this research provides an approach for the FGM characterization method based on acoustic microscopy.

## Figures and Tables

**Figure 1 sensors-22-04052-f001:**
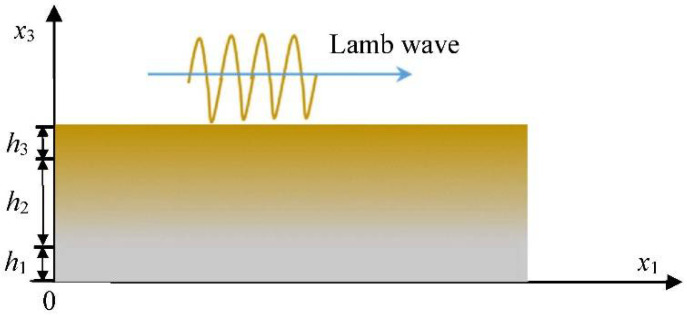
Schematic diagram of Lamb wave propagation and spatial coordinate system in a functionally graded material sandwich panel.

**Figure 2 sensors-22-04052-f002:**
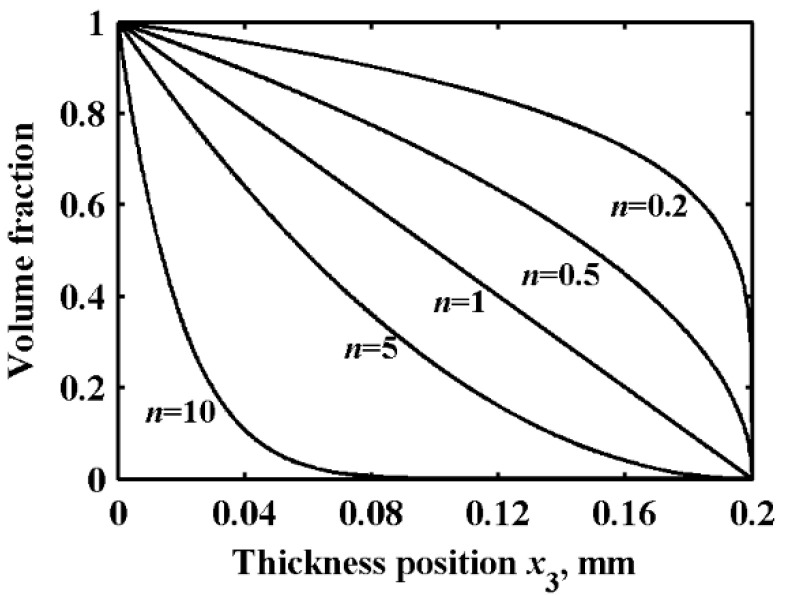
Volume fraction distribution of copper.

**Figure 3 sensors-22-04052-f003:**
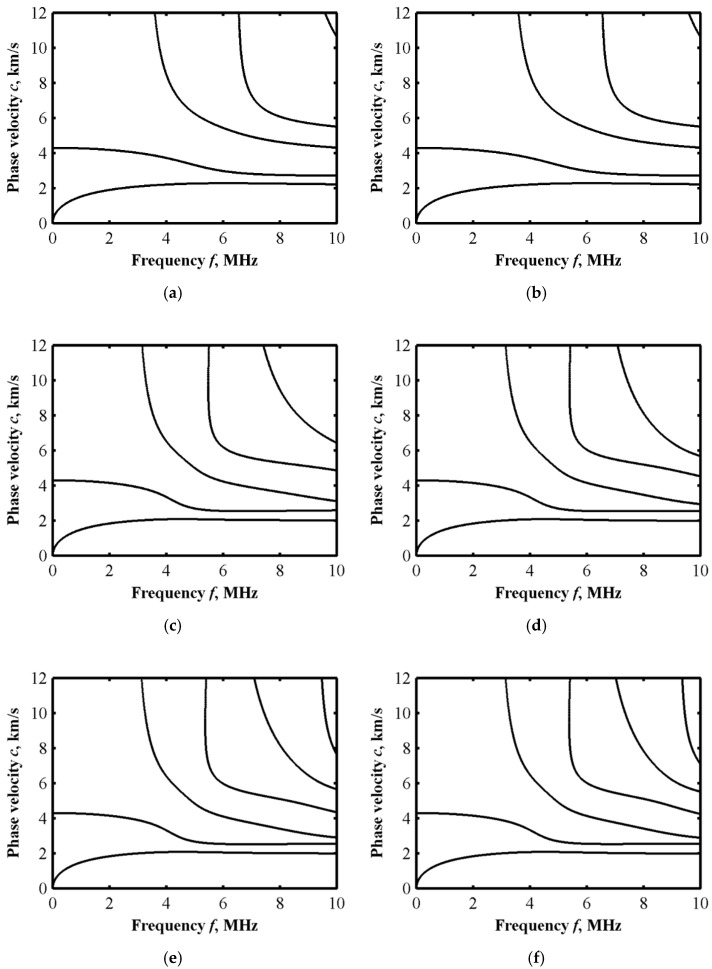
Lamb wave dispersion curves under different cutoff orders: (**a**) *M* = 3; (**b**) *M* = 4; (**c**) *M* = 5; (**d**) *M* = 6; (**e**) *M* = 7; (**f**) *M* = 8; (**g**) *M* = 9; (**h**) *M* = 7, 8, 9 comparison.

**Figure 4 sensors-22-04052-f004:**
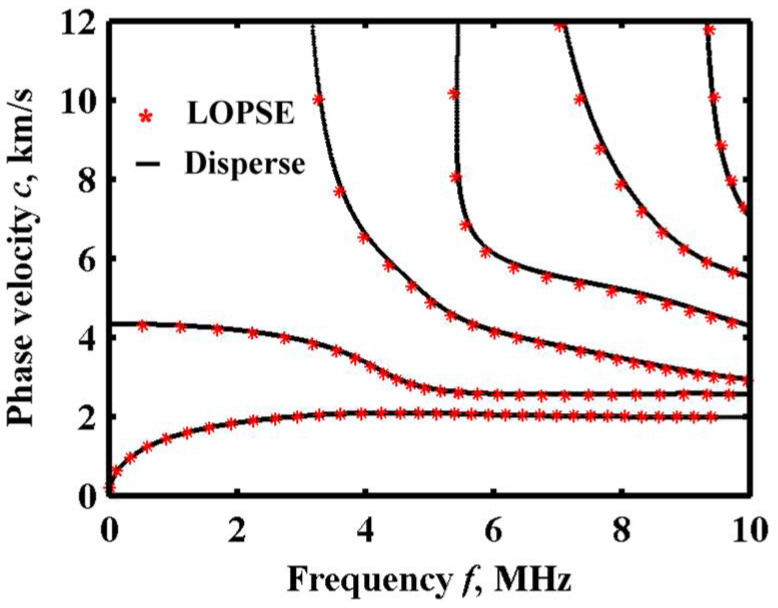
Lamb wave dispersion curve with cutoff order *M* = 8.

**Figure 5 sensors-22-04052-f005:**
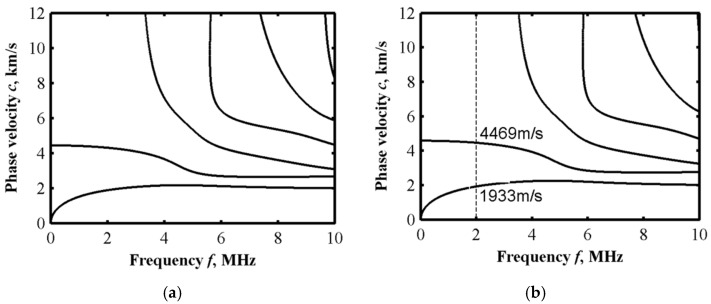
Lamb wave dispersion curve in FGM sandwich plates. (**a**) *n* = 0.5; (**b**) *n* = 1; (**c**) *n* = 5; (**d**) *n* = 10; (**e**) *n* = 20; (**f**) double-layered plate with 0.1 mm Cu and 0.3 mm steel.

**Figure 6 sensors-22-04052-f006:**
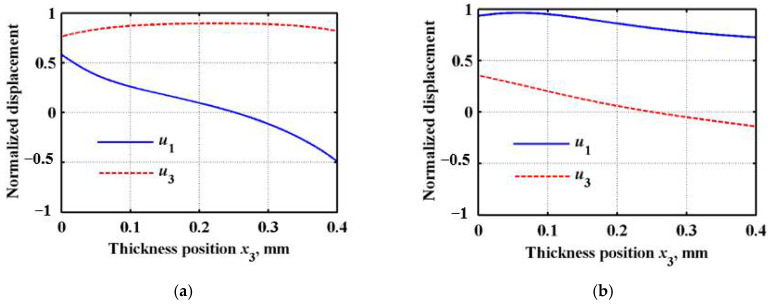
Lamb wave displacement distribution curve in FGM sandwich plate. (**a**) A_0_ mode; (**b**) S_0_ mode.

**Figure 7 sensors-22-04052-f007:**
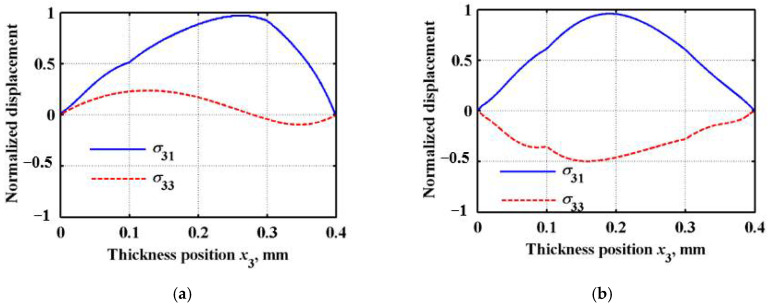
Lamb wave stress distribution curve in FGM sandwich plate. (**a**) A_0_ mode; (**b**) S_0_ mode.

**Figure 8 sensors-22-04052-f008:**
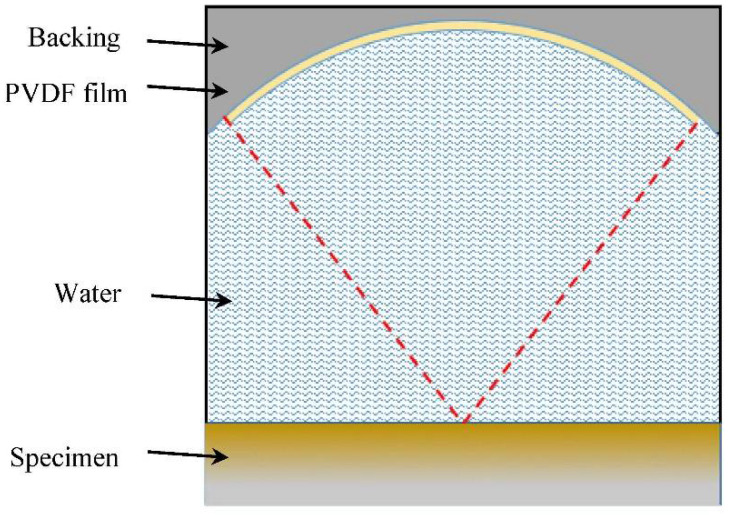
Finite element simulation model.

**Figure 9 sensors-22-04052-f009:**
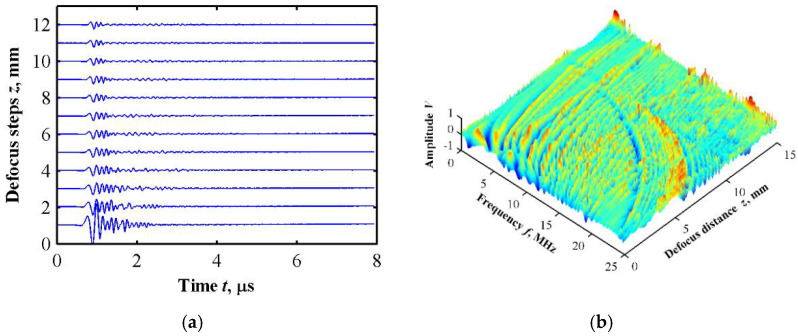
Defocusing experiment simulation results. (**a**) Time domain signal waveform at different defocusing positions; (**b**) frequency domain diagram after time domain Fourier transform; (**c**) frequency peak tracing after spatial Fourier transform; (**d**) Lamb wave dispersion curves.

**Figure 10 sensors-22-04052-f010:**
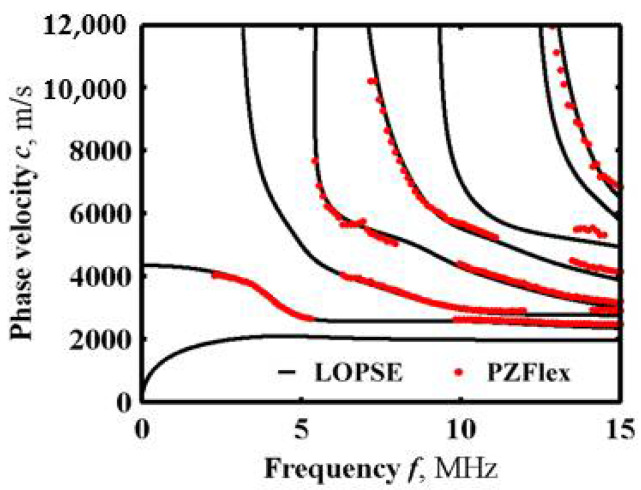
Comparison of simulation results with theoretical results.

**Table 1 sensors-22-04052-t001:** Parameters of Cu and steel [20].

Material	C_11_ (GPa)	C_13_ (GPa)	C_33_ (GPa)	C_55_ (GPa)	*ρ* (kg/m^3^)
Cu	154.8	81.5	154.8	36.7	8292
Steel	275.0	113.2	275.0	80.9	7900

**Table 2 sensors-22-04052-t002:** Parameters of sliced FGM layer.

Layer	C_11_ (GPa)	C_13_ (GPa)	C_33_ (GPa)	C_55_ (GPa)	*ρ* (kg/m^3^)
*N* = 1	154.8	81.5	154.8	36.7	8292.0
*N* = 2	157.6	82.2	157.6	37.7	8282.9
*N* = 3	160.7	83.0	160.7	38.8	8272.8
*N* = 4	164.2	84.0	164.2	40.1	8261.5
*N* = 5	168.1	85.0	168.1	41.6	8248.5
*N* = 6	172.8	86.2	172.8	43.3	8233.3
*N* = 7	178.5	87.7	178.5	45.4	8214.7
*N* = 8	186.0	89.7	186.0	48.2	8190.2
*N* = 9	197.6	92.8	197.6	52.4	8152.6
*N* = 10	275.0	113.2	275.0	80.9	7900.0

**Table 3 sensors-22-04052-t003:** The material property parameters of the model.

Material	Density 𝜌 (Kg/m^3^)	Longitudinal Wave Velocity C_L_ (m/s)	Transverse Wave Velocity C_T_ (m/s)
Back10	2975	1960	1047
PVDF	1780	—	—
Water	1000	1496	—
Cu	8292	4321	2103
FGM layer	8282.9	4357.7	2128.5
	8272.8	4398.4	2156.8
	8261.5	4444.0	2188.5
	8248.5	4496.1	2224.7
	8233.3	4557.4	2267.2
	8214.7	4632.5	2319.4
	8190.2	4731.2	2388.0
	8152.6	4882.5	2493.1
Steel	7900	5900	3200

## Data Availability

Not applicable.

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
