# Peer review of "Lamb Waves Propagation Characteristics in Functionally Graded Sandwich Plates"

_sensors, 2022, doi:10.3390/s22114052_

Round 1
Reviewer 1 Report
The authors investigate the propagation of Lamb waves using a Legendre polynomial expansion series. This provides a new perspective to a well-studied problem.
The following comments are relevant:
1) There are several basic misconceptions in the paper. For example, it is stated that “the finite element method is an analytical method”. This is not correct, the finite element method is a numerical method.
2) Papers with multiple co-authors are wrongly referred. It should not be “Cheng [15]” but “Cheng et al. [15]”.
3) No reference is provided for the data in Table 2. (What is “Disperse”? A reference should be provided)
4) It is unclear why the authors choose to use a layered (homogenous) approach for the finite element analysis. The authors should clearly state that they are adopting a “homogenous” approach (versus a “graded” one), explain the reasons why and provide context referring to the literature (see, e.g., Materials 2019, 12(2), 287 and J. Appl. Mech. 2000, 67, 819–822)
5) The paper is poorly written. The figures are of low quality and there are numerous typos (e.g., “Labaratory” on page 7)
Author Response
Dear editor and reviewer:
Thank you very much for your comments about our manuscript submitted to SENSORS (Lamb Waves Propagation Characteristics in Functionally Graded Sandwich Plates, sensors-1729263). We have checked the manuscript and revised it according to the new comments carefully. Also, all the modified expressions are red marked in the revised manuscript.

Reviewer 2 Report
Please see the attachment.

Author Response
Dear editor and reviewer:
Thank you very much for your comments about our manuscript submitted to SENSORS (Lamb Waves Propagation Characteristics in Functionally Graded Sandwich Plates, sensors-1729263). We have checked the manuscript and revised it according to the new comments carefully. Also, all the modified expressions are red marked in the revised manuscript

Reviewer 3 Report
1. There are some typo mistakes in the draft; should be corrected. such as in line 286, Techneque. 2. In line 229, S0 and A0 modes are mentioned without giving their full name. Are they symmetrical and antisymmetric zero-order mode? If so, please mention it. 3. What happens to the higher order modes when the frequency is increased?Author Response
Dear editor and reviewer:
Thank you very much for your comments about our manuscript submitted to SENSORS (Lamb Waves Propagation Characteristics in Functionally Graded Sandwich Plates, sensors-1729263). We have checked the manuscript and revised it according to the new comments carefully. Also, all the modified expressions are red marked in the revised manuscript.

Round 2
Reviewer 1 Report
The authors have tried to address all my concerns. Publication can now be recommended.
Reviewer 2 Report
The revised version may be accepted for publication.